# Cavity electromechanics with parametric mechanical driving

D. Bothner[1✉], S. Yanai[1], A. Iniguez-Rabago [1], M. Yuan[1,2], Ya.M. Blanter[1] & G.A. Steele[1✉]

Microwave optomechanical circuits have been demonstrated to be powerful tools for both exploring fundamental physics of macroscopic mechanical oscillators, as well as being promising candidates for on-chip quantum-limited microwave devices. In most experiments so far, the mechanical oscillator is either used as a passive element and its displacement is detected using the superconducting cavity, or manipulated by intracavity fields. Here, we explore the possibility to directly and parametrically manipulate the mechanical nanobeam resonator of a cavity electromechanical system, which provides additional functionality to the toolbox of microwave optomechanics. In addition to using the cavity as an interferometer to detect parametrically modulated mechanical displacement and squeezed thermomechanical motion, we demonstrate that this approach can realize a phase-sensitive parametric amplifier for intracavity microwave photons. Future perspectives of optomechanical systems with a parametrically driven mechanical oscillator include exotic bath engineering with negative effective photon temperatures, or systems with enhanced optomechanical nonlinearities.

[1] Kavli Institute of Nanoscience, Delft University of Technology, PO Box 5046, 2600 GA Delft, The Netherlands. [2] Present address: Paul-Drude-Institut für Festkörperphysik Leibniz-Institut im Forschungsverbund Berlin e.V., Hausvogteiplatz 5-7, 10117 Berlin, Germany. ✉email: daniel.bothner@gmail.com; g.a.steele@tudelft.nl

Superconducting microwave circuits have been demonstrated to be extremely powerful tools for the fields of quantum information processing[1–3], circuit quantum electrodynamics[4–8], astrophysical detector technologies[9] and microwave optomechanics[10–12]. In the latter, microwave fields in superconducting cavities are parametrically coupled to mechanical elements such as suspended capacitor drumheads or metallized nanobeams, enabling high-precision detection and manipulation of mechanical motion. Milestones achieved in the field include sideband-cooling of mechanical oscillators to the quantum ground state[11], strong coupling between photons and phonons[13], the generation of non-Gaussian states of motion[14–16] or the entanglement between two mechanical oscillators[17].

Recently, there are increasing efforts taken towards building passive and active quantum-limited microwave elements for quantum technologies based on microwave optomechanical circuits, connecting the fields of microwave optomechanics, circuit quantum electrodynamics and quantum information science[18–20]. Among the most important developments into this direction are the demonstration of microwave amplification by blue sideband driving in simple optomechanical circuits[21], and the realization of directional microwave amplifiers[22] as well as microwave circulators[23,24] in more complex multimode systems[25].

Recent theoretical work[26–28] on optomechanical systems with a parametrically driven mechanical oscillator proposed the use of mechanical parametric driving to enable parametric amplification with enhanced bandwidth and reduced added noise, compared to the case of an optomechanical amplifier using a blue-sideband drive[26]. Furthermore, the authors predict that there is a parameter regime that results in an effective density of states, which can be interpreted as an effective negative temperature for cavity photons[26]. Other related recent works have predicted enhancements of the optomechanical coupling[27] and the generation of non-Gaussian microwave states[28]. Direct electrostatic driving of a mechanical element in an microwave electromechanical cavity using a combination of DC fields and electrical fields resonant with the lower frequency mechanical device have been used in the past for probing mechanical resonators in cavity devices[10,29,30]. These schemes also allow tuning of the mechanical frequency in an optomechanical cavity[29–31] and enable direct parametric driving of the mechanical resonator[32]. Using this electrostatic tuning for parametric driving in an electromechanical system, however, has until now not been explored.

Here, we present measurements of a superconducting microwave optomechanical device in which we use direct electrostatic driving to achieve strong parametric modulation of the mechanical resonator. By modulating the mechanical resonance frequency, we generate phase-sensitive parametric amplitude amplification and thermomechanical noise squeezing of the mechanical motion, both detected using optomechanical cavity interferometry[10]. Furthermore, we demonstrate how parametric modulation of the mechanical resonance frequency can be used to generate phase-sensitive amplification of a microwave probe tone, which is three orders of magnitude larger in frequency than the parametric pump tone itself. For the operation of the microwave amplifier, the optomechanical system can be driven on the red cavity sideband, which allows for simultaneous mechanical cooling and microwave amplification. The experimental implementation presented here provides an optomechanical platform for further exploration of phase-sensitive quantum-limited amplification and photon bath engineering using mechanical parametric driving.

## Results

**The device.** Figure 1 shows an image of a superconducting coplanar waveguide (CPW) quarter-wavelength ($\lambda/4$) resonator used as a microwave cavity. The cavity is patterned from a

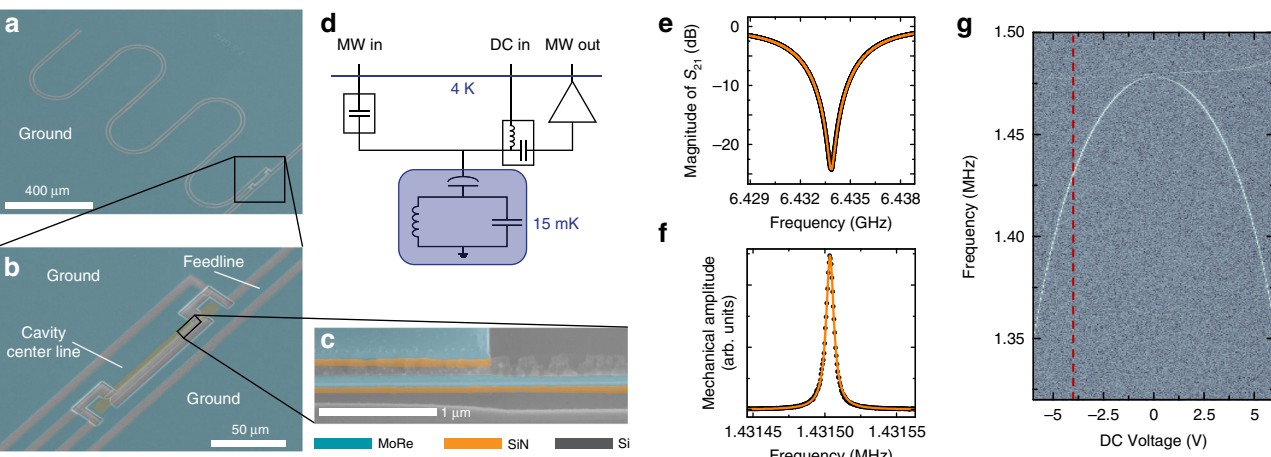

**Fig. 1 Superconducting circuit nano-electromechanical system with electrostatic and low-frequency access. a** False-color scanning electron microscopy image of a superconducting quarter-wavelength cavity (here for $\omega_c = 2\pi \cdot 7.5$ GHz), capacitively side-coupled to a coplanar waveguide feedline. The molybdenum-rhenium (MoRe) metallization is shown in blue and the silicon (Si) substrate in gray. **b** Zoom into the coupling capacitance region, where the mechanical nanobeam as part of the coupling capacitance is visible. The dimensions of the beam, which consists of MoRe on top of high-stress silicon-nitride ($Si_3N_4$), are 100 μm × 150 nm × 143 nm. **c** A magnified view of the suspended nanobeam. **d** Simplified circuit and measurement scheme, showing a lumped element circuit representation of the device as well as the microwave (MW) input and output lines (including a DC block and high electron mobility transistor amplifier shown as triangle) and the DC (directed current) input line connected to the microwave lines via a bias-tee. A more detailed version of the setup is given in Supplementary Note 1. **e** Cavity resonance data (black) and fit curve (orange). From the fit, we extract the cavity resonance frequency $\omega_c = 2\pi \cdot 6.434$ GHz and the internal and external linewidths $\kappa_i = 2\pi \cdot 370$ kHz and $\kappa_e = 2\pi \cdot 5.7$ MHz, respectively. **f** Resonance curve of the mechanical oscillator readout via the superconducting cavity. Data are shown as black dots, a Lorentzian fit as orange line. From the fit we extract the mechanical resonance frequency $\Omega_m = 2\pi \cdot 1.4315$ MHz and a quality factor $Q_m = 195,000$. **g** Optomechanically detected excitation spectrum of the nanobeam vs. applied DC voltage. The bright line resembling an inverted parabola represents the resonance of the in-plane mode, which was used everywhere throughout this paper. The thin second line around 1.48 MHz corresponds to the mechanical out-of-plane mode. The red dashed line at $V_{dc} = -4$ V indicates the voltage operation point we chose to use.

~60-nm-thick film of 60/40 molybdenum-rhenium alloy (MoRe, superconducting transition temperature $T_c \sim 9$ K[33]) on a $10 \times 10$ mm$^2$ and 500 μm-thick high-resistivity silicon substrate; cf. Methods section and Supplementary Fig. 1. For driving and readout, the cavity is capacitively side-coupled to a transmission feedline by means of a coupling capacitance $C_c = 16$ fF. The cavity has a fundamental mode resonance frequency $\omega_c = 2\pi \cdot 6.434$ GHz and internal and external linewidths $\kappa_i = 2\pi \cdot 370$ kHz and $\kappa_e = 2\pi \cdot 5.7$ MHz, respectively. The transmission spectrum of the cavity around its resonance frequency is shown in Fig. 1e; for details on the device modeling and fitting, see Supplementary Note 2.

The superconducting cavity is parametrically coupled to a MoRe-coated high-stress Si$_3$N$_4$ nanobeam, which is electrically integrated into the transmission feedline. The nanobeam has a width $w = 150$ nm, a total thickness $t = 143$ nm (of which ~83 nm are Si$_3$N$_4$ and 60 nm are MoRe) and a length $r = 100$ μm. It is separated from the center conductor of the cavity by a ~200-nm-wide gap (cf. Fig. 1c) and we estimate the electromechanical coupling strength to be $g_0 = 2\pi \cdot 0.9$ Hz. More design and fabrication details are described in the Methods section and Supplementary Fig. 1.

The mechanical nanobeam oscillator has a resonance frequency of its fundamental in-plane mode of $\Omega_{m0} = 2\pi \cdot 1.475$ MHz. It can be significantly tuned by applying a DC voltage $V_{dc}$ between center conductor and ground of the CPW feedline, adding an electrostatic spring constant to the intrinsic spring (cf. Supplementary Note 4). The measured functional dependence of the resonance frequency on DC voltage is shown in Fig. 1g. Throughout this whole article, we bias the mechanical resonator with $V_{dc} = -4$ V, leading to a resonance frequency $\Omega_m = 2\pi \cdot 1.4315$ MHz and a linewidth $\Gamma_m \approx 2\pi \cdot 7.5$ Hz. A resonance curve of the mechanical oscillator at $V_{dc} = -4$ V is shown in Fig. 1f.

The device is operated in a dilution refrigerator with a base temperature of $T_b = 15$ mK, which corresponds to a thermal cavity occupation of $\frac{k_B T_b}{\hbar \omega_c} \sim 0.05$ photons. Assuming the mode temperature of the nanobeam being the fridge base temperature, we expect an average occupation of the mechanical mode with $n_m = \frac{k_B T_m}{\hbar \Omega_m} \sim 220$ thermal phonons.

**Parametric mechanical amplitude amplification.** When the resonance frequency $\Omega_m$ of a harmonic oscillator is modulated with twice the resonance frequency $\Omega_p = 2\Omega_m$, then a small starting amplitude of the oscillator motion can be increased or reduced, depending on the relative phase between the oscillator motion and the frequency modulation[34,35]. To modulate the resonance frequency of a mechanical oscillator, one of the relevant system parameters like the oscillator mass $m$ or the restoring spring force constant $k$ can be modulated. Here, we follow the latter approach and modulate the effective spring constant of the nanobeam by applying a combination of a static voltage $V_{dc}$ and an oscillating voltage $V_{2\Omega} \cdot \sin 2\Omega t$ with roughly twice the mechanical resonance frequency $\Omega \sim \Omega_m$. The static voltage adds an electrostatic spring contribution $k_{dc}$ to the intrinsic spring constant $k_m$ and the oscillating part modulates the total spring constant with ~$2\Omega_m$. The capacitive modulation of the mechanical resonance frequency is a natural choice for superconducting cavity electromechanics[36], but other possibilities have been explored as well, mainly in the optical domain with nonmetallized mechanical oscillators. It has been demonstrated that the time-varying dynamical backaction of modulated laser beams[37,38] and switching of trapping frequencies for levitated dielectric particles[39] can also be utilized for mechanical parametric amplification.

In addition to the parametric driving, we slightly excite the mechanical oscillator by adding a near-resonant oscillating

voltage $V_0 \cos(\Omega t + \phi_p)$ and characterize its steady-state displacement amplitude depending on the parametric modulation amplitude $V_{2\Omega}$ and on the relative phase difference between resonant drive and parametric modulation $\phi_p$. The mechanical amplitude is detected by monitoring the optomechanically generated sidebands to a microwave drive tone sent into the cavity, which is constant in amplitude and frequency with $\omega \sim \omega_c$ (cf. Fig. 2a).

We operate the nanobeam in a regime of voltages where it can be modeled by the equation of motion

$$\ddot{x} + \Gamma_m \dot{x} + \frac{1}{m}\left[k_0 + k_p \sin 2\Omega t\right]x = \frac{F_0}{m}\cos\left(\Omega t + \phi_p\right), \quad (1)$$

where $m$ is the effective nanowire mass, $x$ is the effective nanowire displacement, $k_0 = k_m + k_{dc}$, $k_p \propto V_{dc}V_{2\Omega}$ and $F_0 \propto V_{dc}V_0$. From an approximate solution of this equation of motion, the parametric amplitude gain $G_p = |x|_{on}/|x|_{off}$ can be derived to be given by

$$G_p = \left[\frac{\cos^2(\phi_p + \varphi)}{\left(1 + \frac{V_{2\Omega}}{V_t}\right)^2} + \frac{\sin^2(\phi_p + \varphi)}{\left(1 - \frac{V_{2\Omega}}{V_t}\right)^2}\right]^{1/2}. \quad (2)$$

The detuning-dependent threshold voltage $V_t$ for parametric instability in this relation is given by

$$V_t = V_{t0}\sqrt{1 + \frac{4\Delta_m^2}{\Gamma_m^2}} \quad (3)$$

with the threshold voltage on resonance $V_{t0}$ and the detuning from mechanical resonance $\Delta_m = \Omega - \Omega_m$. The phase $\varphi = -\arctan(2\Delta_m/\Gamma_m)$ considers the detuning-dependent phase difference between the near-resonant driving force and the mechanical motion. Details on the theoretical treatment of the device are given in Supplementary Note 6.

Figure 2 summarizes our results on the phase and detuning-dependent parametric frequency modulation. When we excite the mechanical resonator exactly on resonance, apply a parametric modulation with twice the resonance frequency and sweep the phase $\phi_p$, we find an oscillatory behavior between amplitude amplification and de-amplification with a periodicity of $\Delta\phi_p = \pi$ (cf. Fig. 2b). To explore the dependence of the amplification on the parametric modulation amplitude $V_{2\Omega}$, we repeat this experiment for different voltages $V_{2\Omega}$ and extract maximum and minimum gain by fitting the data with Eq. (2) for $V_t = V_{t0}$ and $\varphi = 0$. The extracted values follow closely the theoretical curves up to a voltage $V_{2\Omega} \approx 0.9V_{t0}$, above which we are limited by resonance frequency fluctuations of the mechanical resonator. The maximum gain we achieve by this is about ~22 dB.

In order to characterize the device response also for drive frequencies detuned from resonance, we repeat the above measurements for different detunings and extract the maximum and minimum gain for each of these data sets. Hereby, we always keep the parametric drive frequency twice the excitation frequency and not twice the resonance frequency. The maximum and minimum values of gain we find for $V_{2\Omega} \approx 0.75V_{t0}$ are shown in Fig. 2d and are in good agreement with theoretical curves shown as lines. We note that the dependence of maximum and minimum gain of detuning is not Lorentzian lineshaped, as the threshold voltage is detuning dependent itself and the deviations between experimental data and theoretical lines mainly occur due to slow and small resonance frequency drifts of the nanobeam. Moreover, the phase between near-resonant excitation drive and parametric modulation for maximum/minimum gain does not have a constant value; it follows an arctan-function as is discussed in more detail in Supplementary Note 6.

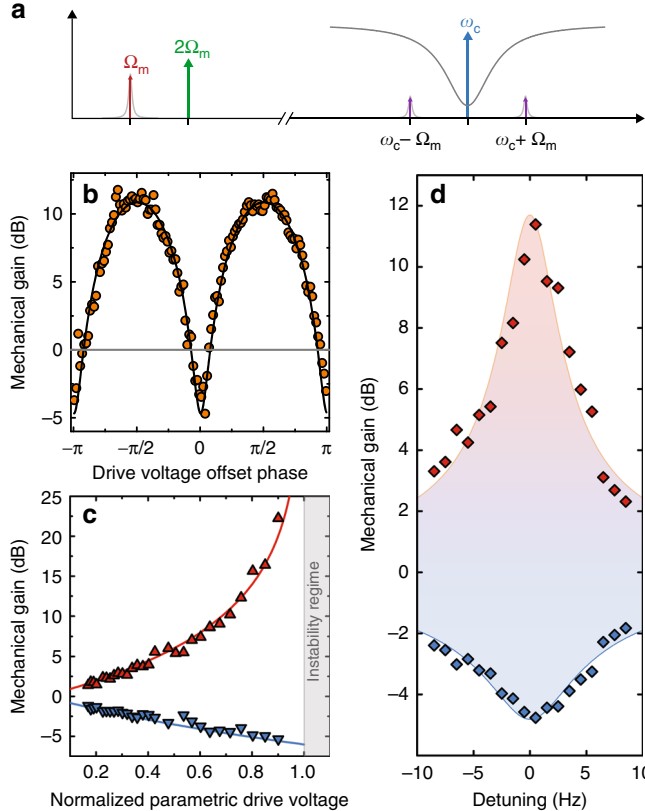

**Fig. 2 Electromechanical detection of parametric, phase-sensitive mechanical amplitude amplification. a** Experimental scheme. The mechanical oscillator is coherently driven by a combination of DC and alternating voltage with frequency $\Omega \sim \Omega_m$, while the electrostatic spring constant is modulated with twice this frequency $2\Omega \sim 2\Omega_m$. Via the optomechanical coupling, the mechanical oscillations generate sidebands to a microwave pump tone sent to the cavity with frequency $\omega = \omega_c$, which are used for homodyne detection of the mechanical amplitude. **b** Mechanical amplitude gain $20\log_{10} G_p$ vs. offset phase $\phi_p$ between resonant drive and parametric modulation. When the phase is swept, the amplitude is oscillating between amplification or de-amplification with a periodicity of $\pi$. Circles show data and the line shows a fit with the theoretical expression Eq. (2). **c** Maximum and minimum gain on resonance vs. parametric modulation strength. The maximum ($\phi_p = \pi/2$) and minimum ($\phi_p = 0$) gain values on resonance follow the theoretical curves (lines) up to a maximum gain of ~22 dB. For stronger parametric modulation amplitudes close to the instability threshold (indicated as vertical line), the gain in our experiments is limited by resonance frequency fluctuations of the mechanical resonator. **d** Maximum and minimum gain vs. detuning from resonance. For a driving frequency slightly detuned from resonance, the maximum gain gets reduced compared to the resonant case. Points are extracted from phase-sweep curve fits. Lines show the corresponding theoretical curves and the shaded area contains all gain values achievable by changing $\phi_p$.

In summary, we have achieved an excellent experimental control and theoretical modeling regarding the parametric amplification of the coherently driven nanobeam in both parameters, the relative phase between the drives and the detuning from mechanical resonance.

**Thermomechanical noise squeezing.** Due to a large residual occupation of the mechanical mode with $10^2$–$10^3$ thermal phonons, its displacement is subject to thermal fluctuations, which in

a narrow bandwidth can be described by[34]

$$x_{th}(t) = X(t)\cos\Omega_m t + Y(t)\sin\Omega_m t. \quad (4)$$

Here, $X(t)$ and $Y(t)$ are random variable quadrature amplitudes, which vary slowly compared to $\Omega_m^{-1}$. Similarly to the coherently driven mechanical amplitude detection discussed above, this thermal motion or thermomechanical noise can be measured by optomechanical sideband generation in the output field of a microwave signal sent into the superconducting cavity (cf. the inset schematic in Fig. 3a).

We measure the thermomechanical noise quadratures $X(t)$ and $Y(t)$ with and without parametric pump. An exemplary result is shown in Fig. 3b. As we have demonstrated above by amplification and de-amplification of a coherent excitation, one of the quadrature amplitudes, here $Y(t)$, is getting amplified while the other, here $X(t)$, is simultaneously reduced, when the mechanical resonance frequency is parametrically modulated with $2\Omega_m$. This puts the mechanical nanobeam into a squeezed thermal state. From the time traces of the quadratures, we reconstruct by means of a Fourier transform the power spectral density (PSD) of the noise as shown in Fig. 3a. With parametric driving, the total PSD is larger than without, in particular close to $\Omega_m$, as the additional energy pumped into the amplified quadrature $Y(t)$ is larger than the energy reduction in $X(t)$ and at the same time the total linewidth decreases for the same reason.

From the time traces, we can also generate quadrature amplitude histograms, shown in the bottom panels of Fig. 3b. In the histograms the squeezing of the thermal noise is apparent as a deformation from a circular, two-dimensional (2D) Gaussian distribution in the case without parametric pump to a cigar-like-shaped overall probability distribution, when the parametric modulation is applied. To determine the squeezing factor we achieve by this, we integrate the 2D-histograms along the $Y$-quadrature and extract the variance $\sigma_X^2$ of the $X$-quadrature from a Gaussian fit to the resulting data (cf. Fig. 3c). Analogously, we obtain the variance $\sigma_Y^2$ for the $Y$-quadrature. To calibrate out the noise of the HEMT amplifier, we substract the independently measured variance of the amplifier noise $\sigma_{amp}^2$ and define the bare variances $\Delta X^2 = \sigma_X^2 - \sigma_{amp,X}^2$ and $\Delta Y^2 = \sigma_Y^2 - \sigma_{amp,Y}^2$.

For the parametric modulation amplitude $V_{2\Omega}/V_t \approx 0.67$ used here, we find the squeezing factor

$$s = \frac{\Delta X_{on}^2}{\Delta X_{off}^2} = 0.49, \quad (5)$$

where $\Delta X_{on}^2$ and $\Delta X_{off}^2$ are the $X$-quadrature variances with the parametric drive on and off, respectively. This squeezing factor is below the usually mentioned 3 dB limit due to the finite analysis bandwidth (cf. discussion in Supplementary Note 7). Using more advanced squeezing schemes with feedback[40–42] or based on measurement[43,44], it has been demonstrated that the variance of the $X$-quadrature can be squeezed by even more than 3 dB, but these approaches typically operate in the instability regime and suppress the corresponding amplification of the $Y$-quadrature. The variance $\sigma_{amp}^2$ is the quadrature noise originating from the cryogenic amplifier in our detection chain and is measured by monitoring the noise slightly detuned from the mechanical resonance.

From an analysis of the individual quadrature power spectral densities and variances, based on ref. [41], we estimate the effective temperatures of the quadratures to be increased by about 18% for $X$ and about 40% for $Y$ due to the parametric drive. We believe that this excess noise as compared to an ideal parametric amplifier is induced by resonance frequency fluctuations of the mechanical oscillator and could be reduced by a device with higher frequency stability.

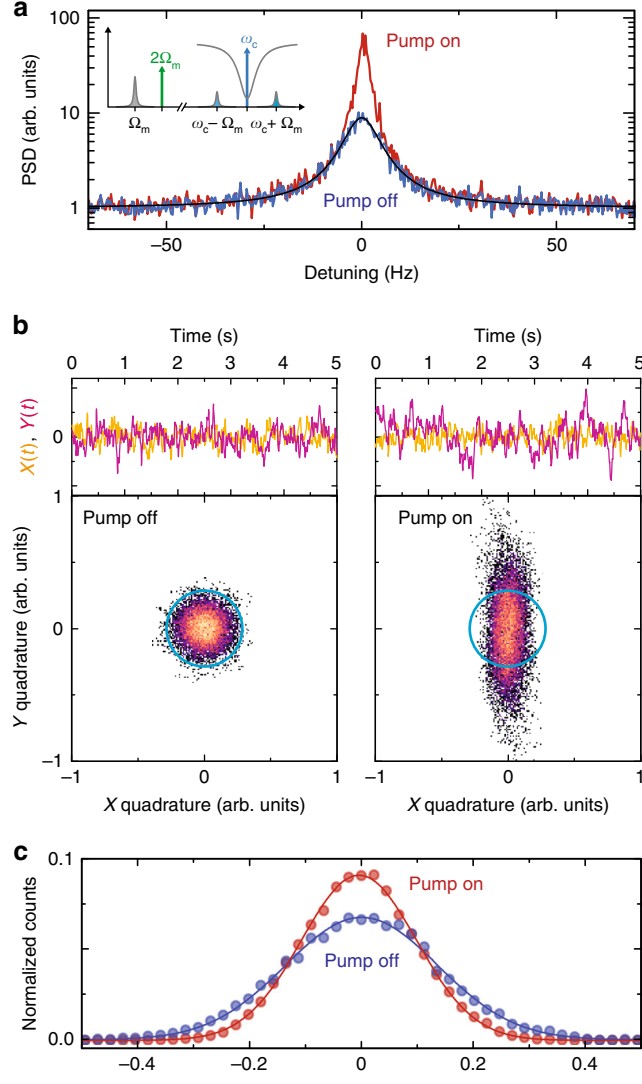

**Fig. 3 Interferometric detection of squeezed thermomechanical noise in a nanomechanical oscillator. a** The thermal displacement fluctuations generate sidebands at $\omega = \omega_c + \Omega_m$ and $\omega = \omega_c - \Omega_m$ to a microwave tone sent to the cavity at $\omega = \omega_c$ as schematically shown in the inset. After down-conversion, we detect these sidebands and the corresponding power spectral density is shown for the parametric modulation switched off as blue line and with the parametric modulation switched on as red line. The black line is a Lorentzian fit to the data without parametric modulation. **b** shows the quadratures of the thermal displacement fluctuations vs. time in the top panels and as histograms (taken for 300 s of measurement time) in the bottom panels. Without parametric modulation, the thermal fluctuations are distributed equally in both quadratures (left side) and the quadrature histogram is a rotational symmetric Gaussian curve; with a parametric modulation applied, as shown on the right side, the fluctuations in one quadrature get amplified while the fluctuations in the second quadrature get de-amplified. The result is a squeezed thermal state. The colorscale represents histogram counts from low (dark) to high (orange) values. White pixels correspond to no recorded counts. The blue circles in the histogram plots are guides to the eye. In (**c**) we plot the distribution of X-quadrature values for the histograms shown in (**b**) as dots and Gaussian fits as lines. When the parametric modulation is switched on, the variance of the X-quadrature gets significantly decreased and the squeezing factor is approximately $s = 0.49$. The histograms are normalized to the total number of ~13,000 data points. Data were taken for $V_{2\Omega}/V_t = 0.67$.

**Parametric microwave amplification.** In a cavity optomechanical system, the mechanical oscillator can not only be coherently driven by a directly applied resonant force, but also by amplitude modulations of the intracavity field. Such a near-resonant amplitude modulation can be generated by sending two microwave tones with a frequency difference close to the mechanical resonance into the cavity. Here, we apply a strong microwave drive tone on the red sideband of the cavity, i.e., at $\omega_d = \omega_c - \Omega_m$, and add a small probe signal around the cavity resonance frequency at $\omega_p \sim \omega_c$. This experimental scheme generates a phenomenon called optomechanically induced transparency (OMIT), where by interference a narrow transparency window opens up in the center of the cavity absorption dip[45,46]. The width of the transparency window is given by the sum of intrinsic mechanical linewidth $\Gamma_m$ and the additional linewidth due to the red-sideband-drive-induced optical damping $\Gamma_o$. The effect of OMIT effect can be understood as follows. The amplitude beating between the two microwave tones coherently drives the nanobeam by an oscillating radiation pressure force, which transfers energy from the cavity field to the nanobeam. The resulting mechanical motion with frequency $\Omega = \omega_p - \omega_d$ modulates the cavity resonance frequency and hereby generates sidebands to the intracavity drive tone at $\omega_d \pm \Omega$, with a well-defined phase relation to the probe tone. The sideband generated at $\omega_d + \Omega$ interferes with the probe signal and generates OMIT (cf. Fig. 4a for vanishing parametric modulation and Fig. 4b). In Fig. 4b, the transparency window can be seen in the center of the cavity transmission spectrum as extremely narrow spectral line and a zoom into this region, shown in Fig. 4c, reveals the Lorentzian lineshape with a width $\Gamma_{eff} \approx 2\pi \cdot 12$ Hz.

When we perform the OMIT protocol with a parametric modulation applied to the nanobeam, the mechanical oscillations get modified according to the previously shown results, i.e., dependent on the relative phase between the cavity field-induced mechanical oscillation and the parametric modulation, the mechanical amplitude gets amplified or de-amplified. By choosing the optimal phase for each detuning $\Delta_m = \Omega - \Omega_m$, the transparency window amplitude can be increased to values above 1, i.e., the microwave probe tone is amplified by parametrically pumping the mechanical resonator, which is three orders of magnitude smaller in frequency than the probe signal (cf. Fig. 4c). With an amplified mechanical motion, the motion-induced sideband of the drive tone gets amplified as well, such that the total cavity output field at the probe frequency can be enhanced to values larger than 1. Here, we achieve an intracavity field gain of about 14 dB, which corresponds to a net gain of about 7 dB due to the unamplified OMIT signal being significantly below unity transmission. A schematic of OMIT and the amplification mechanism is shown in Fig. 4a.

The observed microwave amplification is, similarly to the bare mechanical amplitude gain, phase-sensitive and modulates between amplification and de-amplification when sweeping the phase of the parametric drive, with a periodicity of $2\pi$. This phase-sensitivity of the microwave gain is shown in Fig. 4d for three different detunings from the mechanical resonance. We note that the phase periodicity here is equivalent to the case of the mechanical amplitude amplification, but due to the details of our theoretical analysis of the system (see Supplementary Note 8) the phase is given for the parametric drive instead of the resonant force here, which doubles its value.

Similar to the mechanical amplitude amplification, the microwave gain depends on the parametric drive voltage, which has a threshold value above which the parametric instability regime begins. When we plot the maximally achievable transmission $|S_{21}|$ exactly on the mechanical resonance vs. the parametric excitation voltage, we find a monotonously increasing behavior as shown in Fig. 4e for three different red-sideband drive powers. Shown are

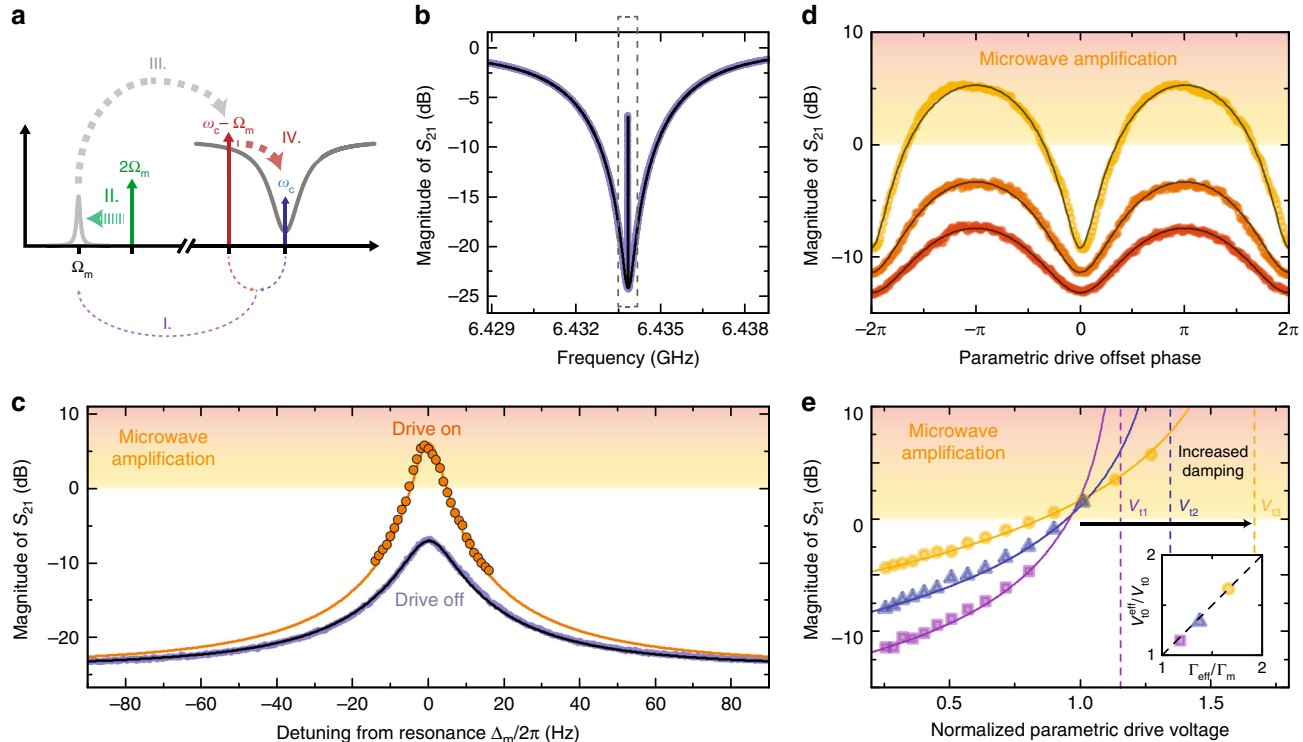

**Fig. 4 Phase-sensitive and tunable microwave amplification by parametric mechanical driving. a** Experimental scheme. The cavity is coherently driven on the red sideband $\omega_d = \omega_c - \Omega_m$. In addition, a small probe tone is swept through the cavity resonance. At the same time, the resonance frequency of the mechanical oscillator is parametrically modulated with $2\Omega$. **b** Optomechanically induced transparency (OMIT) without parametric modulation $V_{2\Omega} = 0$. Data are shown in blue, black line is a fit, the dashed box indicates the zoom-in region shown in panel (**c**). In addition to the data without parametric modulation, we show the highest achieved transmission with parametric driving as orange circles. Close to the mechanical resonance we observe intracavity gain of the probe signal up to ~14 dB and net transmission gain of ~7 dB. The orange line shows a theoretical curve calculated with independently obtained parameters. The schematic in (**a**) visualizes the amplification mechanism. By the beating of the two cavity tones, energy from the cavity field is converted into mechanical motion, which is amplified by parametric modulation. The hereby increased energy is upconverted back to the probe tone frequency as sideband of the red-detuned drive tone. **d** The microwave gain is phase-sensitive; it depends on the phase between the parametric modulation and the intracavity amplitude beating. The three data sets (black lines are fits) show the gain for different detunings from $\omega_p - \omega_d = \Omega_m$ (0 Hz, $2\pi \cdot 7$ Hz and $2\pi \cdot 12$ Hz). **e** Probe-tone gain vs. parametric drive voltage for three different red-sideband drive powers. The parametric drive voltage is normalized to its value obtained in Fig. 2 using a resonant drive for amplitude detection. Lines are calculations based on independently extracted parameters. The parametric instability threshold, indicated by dashed vertical lines, is shifted to higher values with increasing red-sideband drive power, partly due to optical damping, partly due to a power-dependent intrinsic mechanical damping rate. The inset shows the extracted threshold voltage vs. effective mechanical linewidth and as dashed line the theoretical prediction. The cooperativity for (**b**–**d**) is $\mathcal{C} \approx 0.5$ and $\mathcal{C} \approx 0.16, 0.28, 0.5$ for (**e**).

data for drive powers corresponding to cooperativities $\mathcal{C}_1 \sim 0.16$, $\mathcal{C}_2 \sim 0.28$ and $\mathcal{C}_3 \sim 0.5$ or intracavity photon numbers $n_{c1} = 2.25 \cdot 10^6$, $n_{c2} = 4.5 \cdot 10^6$, and $n_{c3} = 9 \cdot 10^6$. The functional dependence of the maximum transmitted power is formally identical to the case without parametric driving

$$|S_{21}|^2 = \frac{\kappa_i^2}{\kappa^2} + \mathcal{C}_p \frac{\Gamma_m}{\Gamma_{eff}^2} \left[ 2 \frac{\kappa_i \kappa_e}{\kappa^2} \Gamma_{eff} + \frac{\kappa_e^2}{\kappa^2} \mathcal{C}_p \Gamma_m \right] \quad (6)$$

with a parametrically enhanced cooperativity

$$\mathcal{C}_p = \frac{\mathcal{C}}{1 - \frac{V_{2\Omega}}{V_{t0}^{eff}}}, \quad (7)$$

where the effective threshold voltage is given by $V_{t0}^{eff} = V_{t0} \Gamma_{eff}/\Gamma_m$. From fits to the data, shown as lines, we can extract the instability threshold voltages, indicated as dashed vertical lines and plotted in the inset vs. effective mechanical linewidth. The threshold gets shifted towards higher values due to an increase of mechanical linewidth, which is partly due to the optical spring and partly due to a microwave power-dependent intrinsic linewidth (see Supplementary Note 5). At the same time, the net microwave

gain increases with increasing sideband drive power, as the baseline (the peak height of the transparency window) is shifted up as well and because the gain in this experiment was limited by the mechanical nonlinearity, cf. Supplementary Note 9, which gets less significant for a larger total mechanical linewidth.

So far, our current device is far from being optimized for large gain, large bandwidth and low added noise for several reasons. Due to the small maximally achieved cooperativity of $\mathcal{C} \sim 0.5$, not all intracavity gain is translated to net output gain. At the same time, the cooperativity limits the amplification bandwidth, which is given by the effective mechanical linewidth $\Gamma_{eff}$. Finally, it would be desirable to operate in the sideband-resolved limit $\Omega_m > \kappa$ to enable ground-state cooling in contrast to the slightly bad cavity limit in our present device, where even for large cooperativities the lowest possible phonon occupation is given by $n_{min} = \frac{\kappa}{4\Omega_m} \approx 1$. The residual phonon occupation, however, directly translates to input-referenced added noise[26]. In an optimized device, operated in the resolved sideband regime and with cooperativities $\mathcal{C} > n_{th}$, all intracavity field gain corresponds to net microwave gain, the bandwidth will be increased by several orders of magnitude compared to the current value and the

amplifier will be near-quantum limited, as has been extensively discussed in ref. [26]. The most straightforward way to achieve these realistic numbers would be a significant increase of the single-photon coupling rate $g_0$ by about a factor of ~10 and a simultaneous decrease of the cavity linewidth by ~10 through a weaker coupling to the feedline (cf. also Supplementary Note 10).

In terms of amplifier performance, such an optimized device would be on par with other, recently developed multimode or multitone optomechanical amplification schemes[20,22,25,47], but provides a simplified setup as it does not require multiple circuit modes or frequency conversion. In contrast, however, to most previously realized optomechanical amplification schemes[21,22,25,48], our system provides phase-sensitive amplification, enabling for example mechanically mediated squeezing of microwave signals[36]. Compared to other cavity-based parametric amplifiers such as Josephson-based circuits[49,50], optomechanical amplifiers suffer from a reduced bandwidth, but with the benefit of a large dynamic range[25]. Considering the additional possibilities arising from the presented scheme such as enhancing optomechanical nonlinearities[27], photon bath engineering[26] and force sensing in hybrid devices with a Bose–Einstein condensate[51–53], our platform offers rich and exciting perspectives for quantum-limited optomechanical device engineering.

## Discussion

In this work, we have demonstrated an electromechanical cavity with mechanical parametric driving. By means of an optomechanical, interferometric readout scheme of a high-quality factor mechanical nanobeam oscillator, we have demonstrated phase-sensitive mechanical amplitude amplification, and observed thermomechanical noise squeezing. We demonstrated that this parametric mechanical drive can be used to implement a phase-sensitive microwave amplification, in a regime where dynamical backaction can simultaneously cool the mechanical resonator. Using the presented experimental platform in an optimized device, it should be possible to cool the mechanical oscillator into its quantum ground state and perform a near-quantum-limited amplification scheme for microwave photons. Furthermore, this approach will allow to explore exotic regimes of bath engineering for microwave cavities[26] and enable other applications of mechanical parametric driving and mechanical squeezing, that have been proposed and discussed in the recent years[27,28].

## Methods

**Device fabrication.** The device fabrication starts with the deposition of a 100-nm-thick layer of high-stress $Si_3N_4$ on top of a 500-µm-thick 2-inch silicon wafer by means of low pressure chemical vapour deposition. Afterwards, 60-nm-thick gold markers on a 10-nm chromium adhesion layer were patterned onto the wafer using electron beam lithography (EBL), electron beam evaporation of the metals and lift-off. Then, the wafer was diced into individual $10 \times 10$ mm$^2$ chips, which were used for the subsequent fabrication steps.

By using a three-layer mask (S1813, tungsten and ARN-7700-18), EBL and several reactive ion etching (RIE) steps with $O_2$ and an $SF_6/He$ gas mixture, the $Si_3N_4$ was thinned down everywhere to ~10 nm on the chip surface except for rectangular patches ($124 \times 9$ µm large) around the future locations of the nanobeams. After resist stripping in PRS3000, the remaining ~10 nm of $Si_3N_4$ were removed in a buffered oxide etching step, which also thinned down the $Si_3N_4$ in the rectangular patch areas to ~ 83 nm. This two-step removal of $Si_3N_4$ by dry and wet etching was performed in order to avoid over-etching with RIE into the silicon substrate.

Immediately afterwards, a ~60-nm-thick layer of superconducting molybdenum-rhenium alloy (MoRe, 60/40) was sputtered onto the chip. By means of another three-layer mask (S1813, W, PMMA 950K A6), EBL, $O_2$ and $SF_6/He$ RIE, the microwave structures were patterned into the MoRe layer. The remaining resist was stripped off in PRS3000.

Finally, the nanobeam patterning and release was performed. The pattern definition was done using another three-layer mask (S1813, W, PMMA 950K A6), EBL and RIE. After the MoRe-$Si_3N_4$ bilayer was completely etched by the $SF_6/He$ gas mixture, the etching was continued for several minutes. As we had chosen the RIE parameters to achieve slight lateral etching, the silicon underneath the narrow nanobeam was etched away by this measure and the beam was released from the substrate. After the nanobeam release, the remaining resist was stripped using an $O_2$ plasma.

A simplified schematic of the fabrication is shown in Supplementary Fig. 1, omitting the patterning of the electron beam markers.

**Mechanical amplitude amplification—measurement routine and data processing.** To measure the mechanical amplitude amplification, we sweep the phase between the drive tone and the parametric pump. In order to sweep the phase, we add a small detuning $\delta$ on the order of ~0.1 Hz to the parametric drive tone, i.e., modulate with $2\Omega + \delta$, and measure a time trace of the down-converted cavity sideband signal at $\Omega$. Then, the parametric phase is given by $\phi_P = \delta t + \gamma$ with an arbitrary offset term $\gamma$. For the down-conversion, we send a resonant microwave tone to the cavity and detect the cavity output field at $\Omega$ after mixing it with the drive tone as local oscillator. This protocol provides us with a voltage signal proportional to the mechanical amplitude (cf. also Supplementary Note 3).

We fitted the resulting power curves with

$$f(t) = \alpha_1 \left[ \frac{\cos^2(\alpha_2 t + \alpha_3)}{(1 + \alpha_4)^2} + \frac{\sin^2(\alpha_2 t + \alpha_3)}{(1 - \alpha_4)^2} \right] \tag{8}$$

and fit parameters $\alpha_i$, from where we get

$$G_{min} = \frac{1}{1 + \alpha_4}, G_{max} = \frac{1}{1 - \alpha_4}. \tag{9}$$

Repeating this procedure for different detunings allows to determine the maximum and minimum gain dependent on $\Delta_m$. Finally, we fit the detuning-dependent maximum and minimum gain points with the corresponding theoretical expression

$$f(\Delta_m) = \frac{\beta_1}{\left( \sqrt{1 + \beta_2(\Delta_m - \beta_3)^2} \pm \beta_4 \right)^2}, \tag{10}$$

where $\pm$ is chosen for minimum and maximum gain, respectively, and $\beta_i$ are the fit parameters. By this method, we determine the maximum gain on resonance with higher fidelity than just setting the excitation frequency to the resonance frequency due to small mechanical resonance frequency drifts and fluctuations of unknown origin. We note that ultimately and for parametric excitation voltages close to the threshold voltage, these frequency shifts also limit the observable gain, as it becomes more and more sensitive to frequency fluctuations as can also be seen in Supplementary Fig. 8, where the range of largest gain gets narrower with increased $V_{2\Omega}/V_t$.

**Thermomechanical noise squeezing—measurement routine.** To characterize the thermomechanical noise of the nanobeam, we send a resonant microwave tone to the cavity and detect the cavity output field after mixing it with the drive tone as local oscillator. This down-converts the motional sidebands to the original mechanical frequency, similar to the protocol for mechanical amplitude amplification. To detect the quadratures of the sidebands $X'(t)$ and $Y'(t)$, we measure the voltage with a lock-in amplifier set to the mechanical resonance frequency with a sample rate of 225 samples/s. The total sampling time was 300 s. This measurement scheme was repeated for different parametric modulation strengths of the mechanical resonance frequency $V_{2\Omega}/V_t$, including $V_{2\Omega}/V_t = 0$. To characterize also the (white) background noise floor originating from the detection amplifier chain, we repeat the measurement for a lock-in center frequency sufficiently detuned from the mechanical resonance such that there is no signature of the mechanical thermal noise included.

**Thermomechanical noise squeezing—data processing.** As first step, we manually rotate the measured quadratures $X'(t)$ and $Y'(t)$ by $-\pi/36$ to obtain the amplified and de-amplified quadratures $X(t)$ and $Y(t)$. We calculate the total PSD by $S = |X(\Omega) + iY(\Omega)|^2$, where $X(\Omega)$ and $Y(\Omega)$ are the Fourier transforms of the recorded $X(t)$ and $Y(t)$, respectively. The obtained spectra are smoothed by applying a 100-point bin averaging and the smoothed spectra are divided by the smoothed background noise spectrum to remove the lock-in amplifier filter function. The result is shown in Fig. 3.

The histogram data in Fig. 3 were obtained by first applying a 40-point moving average and plot each fifth datapoint of the resulting dataset into the histograms. The variances were calculated from Gaussian fits to the $Y$-integrated histograms.

**Parametric microwave amplification—measurement routine and data processing.** Both the measurement routine and the data processing are done in full analogy with the mechanical amplitude amplification. Instead of sweeping the phase, we detune the parametric drive tone by ~0.1 Hz from the frequency difference between the sideband drive and the probe tone. Then, we track the transmission of a probe tone vs. time with a network analyzer. The resulting oscillatory transmission curves of the amplitude are fitted with a function as given in Eq. (9), from which we extract maximum and minimum transmission. To normalize the signal, we calculate the nominal complex background value at the corresponding frequency from the cavity fit divide it off.

## Data availability
The data presented in this study will be available on Zenodo with the identifier https://doi.org/10.5281/zenodo.3713207 upon publication of this work.

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

## Acknowledgements
This research was supported by the Netherlands Organisation for Scientific Research (NWO) in the Innovational Research Incentives Scheme—VIDI, project 680-47-526. This project has received funding from the European Research Council (ERC) under the European Union's Horizon 2020 research and innovation program (grant agreement No. 681476—QOMD) and from the European Union's Horizon 2020 research and innovation program under grant agreement No. 732894—HOT.

## Author contributions

A.I.-R., S.Y. and G.A.S. designed and fabricated the device. D.B., S.Y. and M.Y. performed the measurements. D.B., S.Y. and G.A.S. analyzed the data. D.B. and Y.M.B. developed the theoretical treatment. G.A.S. conceived the experiment and supervised the project. D.B. wrote the manuscript with input from G.A.S. All authors discussed the results and the manuscript.

## Competing interest

The authors declare no competing interests.
