## [Peer Review File · Nature Communications]

Reviewers' comments:

Reviewer #1 (Remarks to the Author):

The work by Bothner et al. investigates electromechanical effects in a microwave cavity using parametric driving of a mechanical resonator. In addition to observing parametric (de)amplification of the vibrations of the nanobeam as well thermomechanical squeezing, they observe for the first time phase-sensitive parametric amplification of the microwave cavity photons induced by the parametric mechanical driving, a phenomenon predicted by Clerk and coworkers (ref. 26). Because it allows for simultaneously cooling the mechanics this type of microwave amplifier has the potential for quantum-limited amplification of microwave photons.

The work reported is of excellent quality. The data presented show a high level of control of both the mechanics and the microwaves. The observations are well-corroborated by the observations. The paper is in addition well-presented and well-documented. The findings should be of a broad interest to both the microwave and the optomechanics communities, and, as such, this manuscript would be suitable for publication in Nature Communications in my opinion.

I have a few minor clarifying comments.

First, the mechanical "amplitude" (de)amplification at zero offset is -5dB (fig. 2), but from eq. 2 it shouldn't be less than -3 dB. Are the authors actually showing power gains?

How close to the threshold was the data presented in Fig. 3 taken?

The authors are a bit vague on the thermal occupation of the mechanics without amplification (10^2 - 10^3). Can this be calibrated?

It would be instructive to the reader to have a number for the intracavity photons (corresponding to a given cooperativity) in the main text.

At the very end of the microwave amplification section the authors write that "the gain in this experiment was limited by the mechanical nonlinearity". Maybe the authors could elaborate on this point, if not in the main text, at least in the supplemental material.

Reviewer #2 (Remarks to the Author):

Report NCOMMS-19-26439 for Nature Communications

In this article, the authors demonstrate mechanical parametric modulation in a superconducting electromechanical system. The basic setup consist of a microwave cavity coupled to a mechanical mode of a nanobeam, forming a standard microwave optomechanical device. The novel element here is the modulation of the mechanical mode around twice it's resonant frequency. The authors show that this modulation leads to phase-sensitive amplification of the mechanical amplitude and thermal mechanical noise squeezing. In addition, they demonstrate that the amplification of the mechanical amplitude

combined with red-sideband driving, translates into phase-sensitive amplification of a microwave signal sent in close to the cavity resonance. The latter was theoretically proposed in Ref. 26 of the manuscript.

To my knowledge, the demonstration of mechanical parametric modulation in an electromechanical system is novel and is an interesting expansion of the manipulation schemes in optomechanical systems. However, the actual performance of the devices as a phase-sensitive amplifier is not convincing yet. Quantum-limited amplifiers required for quantum information processing should provide a gain around 20dB to overcome the noise of the following HEMPT amplifier and have an amplification bandwidth in the order of MHz (or even GHz if not based on a cavity architecture). The present setup is far from these demands with 7dB of gain and a bandwidth in the few-Hz regime. Clearly, the bandwidth is a general issue in optomechanical-based amplifier schemes, limited by the narrow mechanical linewidth (even if optical broadened), but large gain has been demonstrated in electromechanical setups, e.g. in Ref. 20,21,25 of the manuscript. Hence, so far I am not convinced that the results presented in the manuscript do warrant publication in Nature Communications. Further comments are:

1. The abstract states: *In contrast to many other microwave amplification schemes using electromechanical circuits, the presented technique allows for simultaneous cooling of the mechanical element, which potentially enables this type of optomechanical microwave amplifier to be quantum-limited.* First of all 'In contrast many microwave amplification schemes...' is a bit of an overstatement here, clearly, the single-(blue)tone amplification setup in Ref. 21 could not suppress the noise coming from the mechanical mode, but the majority of the recent electromechanical amplifier schemes was based on multi-tone driving with the effect of suppressing the mechanical noise, see Ref. 20, 25 of the manuscript and Phys. Rev. Lett. 118, 103601 (2017). This should be re-phrased and set into the right context. Secondly, the authors speak of the possibility of quantum-limitedness of such an amplifier, but they do not perform an actual noise analysis in their manuscript. And the authors speak of back-action cooling in the conclusion without further elaboration. In the ideal case the results of an actual noise measurement would be desirable, but I can understand that this is not straightforwardly done, however, at least a thorough discussion about the noise properties could be included. In terms of the noise analysis it would be good to understand the limitations coming from the fact that the present setup is not in the resolved sideband regime, in contrast to the proposal in Ref. 26.

2. As mentioned above the properties of the amplifier are not convincing in terms of gain and bandwidth. The authors should state this more clearly, set their device into context with other cavity-based amplifiers, and include ideas how to improve their design in the future. In addition, it would be interesting to know why it was not possible to achieve a higher gain value.

3. Otherwise, I mainly have some minor comments:

- Caption Fig.4: *The schematic shown in d visualizes the amplification mechanism, I guess the authors mean the schematic in a?*
- Typo P.7 last sentence: *mechnaical.*

- General comment: the SI are a bit sketchy and not well-rounded formulated, the authors should consider improving these important additional information.

Reviewer #3 (Remarks to the Author):

In this manuscript, the authors report the parametric amplification of a mechanical beam within a microwave nano optomechanical setup via direct electrical modulation of the mechanical spring constant. They provide evidences by measuring the mechanical gain and the mechanical noise spectrum. They finally show how the parametrically amplified mechanical oscillator allows the amplification of a microwave probe beam in an OMIT setup.

The experiment described and the results shown in the manuscript are of great quality and definitively amongst the state of the art. Showing direct parametric amplification of a mechanical quadrature in a nano optomechanical setup is also of great relevance as many schemes have been proposed in this direction and such results have the potential to motivate more ideas along those lines. While mechanical parametric amplification (and even squeezing below the ZPF which is definitively more challenging) has previously been reported, the present setup is, to my knowledge, novel. Finally, the results are convincing and successfully support their conclusions.

I would make two minor suggestions about the manuscript:

1) As mentionner above, I believe in the novelty of the experiment. However I feel that the authors have not given enough credits to previous works where parametric amplification of mechanical elements have been reported. Also, in this context, I would suggest to the authors to point out explicitly the novelty of their work compared to what have been done before.

2) One of the key aspects of amplification is the amount of total noise added to the amplified mode. As shown in the manuscript, the thermal noise is parametrically amplified such that one quadrature is amplified while the other is de-amplified. However, nothing is said about the total added noise (when integrating both quadratures). I believe this is an important quantity to be discussed in order to know if the amplification process is efficient or not. I would suggest to compare the total added noise measured from Fig. 3 to what is expected from an ideal parametric amplifier (which I believe is $n_{amp} = (n_m + 1/2) * \cosh(2r)$ with r the squeezing parameter).

Considering that those two minor points can be easily addressed by the authors, I recommend the manuscript for publication in Nature Communications.

Best,
Marc-Antoine Lemonde

Reply to the reviewer comments and suggestions

Reply to reviewer #1:

Reviewer #1:

"The work by Bothner et al. investigates electromechanical effects in a microwave cavity using parametric driving of a mechanical resonator. In addition to observing parametric (de)amplification of the vibrations of the nanobeam as well thermomechanical squeezing, they observe for the first time phase-sensitive parametric amplification of the microwave cavity photons induced by the parametric mechanical driving, a phenomenon predicted by Clerk and coworkers (ref. 26). Because it allows for simultaneously cooling the mechanics this type of microwave amplifier has the potential for quantum-limited amplification of microwave photons.

The work reported is of excellent quality. The data presented show a high level of control of both the mechanics and the microwaves. The observations are well-corroborated by the observations. The paper is in addition well-presented and well-documented. The findings should be of a broad interest to both the microwave and the optomechanics communities, and, as such, this manuscript would be suitable for publication in Nature Communications in my opinion."

Reply:

We thank the reviewer for their careful reading of our manuscript and their positive assessment and appreciation of our results.

Reviewer #1:

"I have a few minor clarifying comments.

First, the mechanical "amplitude" (de)amplification at zero offset is -5dB (fig. 2), but from eq. 2 it shouldn't be less than -3 dB. Are the authors actually showing power gains?"

Reply:

In agreement with the definition of quantities in units of (deci-)bels, there is no difference in dB when referring to amplitude or power gain. A gain of 20 dB corresponds to an amplitude ratio of 10 or a power ratio of 100. To avoid confusion for the reader, however, we changed the figure labelling and introduce the used conversion $20 \cdot \log(G_p)$ in the revised version of the manuscript in the caption of Fig. 2.

Reviewer #1:

"How close to the threshold was the data presented in Fig. 3 taken?"

Reply:

The data shown in Fig. 3 were taken at around $V_{2\Omega}/V_t = 0.67$. We added the number to the caption of Fig. 3.

Reviewer #1:

"The authors are a bit vague on the thermal occupation of the mechanics without amplification (10^2 - 10^3). Can this be calibrated?"

Reply:

Unfortunately we do not have sufficient data for a precise calibration of the thermal occupation, i.e., for example a measurement of thermal noise vs base temperature. These measurements are

very time consuming and we did not consider an occupation calibration crucial for the results at the time (and unfortunately the sample is no longer available for additional measurements).

Reviewer #1:

"It would be instructive to the reader to have a number for the intracavity photons (corresponding to a given cooperativity) in the main text."

Reply:

We agree with the reviewer. We added the single-photon coupling rate to the main text, the cooperativities to the caption of Fig. 4 as well as the intracavity photon numbers to the main text at the point where the cooperativities for Fig. 4 are discussed.

Reviewer #1:

"At the very end of the microwave amplification section the authors write that "the gain in this experiment was limited by the mechanical nonlinearity". Maybe the authors could elaborate on this point, if not in the main text, at least in the supplemental material."

Reply:

Yes, indeed, in the presented experiment we were in some sense limited by the nonlinearity of the mechanical oscillator. For an illustration, we added additional data and a corresponding discussion to the Supplementary Material, showing what happens when the parametric drive strength is increased further, i.e., above the parametric voltages used in Fig. 4 of the main manuscript.

We also realized though, that our statement was not very precise here. A smaller probe power would have led to a smaller mechanical amplitude, which most probably would have allowed for higher gain values in the linear regime. Unfortunately, the device is not available anymore for further experiments.

Reply to reviewer #2:

Reviewer #2:

"In this article, the authors demonstrate mechanical parametric modulation in a superconducting electromechanical system. The basic setup consist of a microwave cavity coupled to a mechanical mode of a nanobeam, forming a standard microwave optomechanical device. The novel element here is the modulation of the mechanical mode around twice it's resonant frequency. The authors show that this modulation leads to phase-sensitive amplification of the mechanical amplitude and thermal mechanical noise squeezing. In addition, they demonstrate that the amplification of the mechanical amplitude combined with red-sideband driving, translates into phase-sensitive amplification of a microwave signal send in close to the cavity resonance. The latter was theoretically proposed in Ref. 26 of the manuscript.

To my knowledge, the demonstration of mechanical parametric modulation in an electromechanical system is novel and is an interesting expansion of the manipulation schemes in optomechanical systems."

Reply:

We thank the reviewer for acknowledging that our work is novel and interesting.

Reviewer #2:

"However, the actual performance of the devices as a phase-sensitive amplifier is not convincing yet. Quantum-limited amplifiers required for quantum information processing should provide a gain around 20dB to overcome the noise of the following HEMPT amplifier and have a amplification bandwidth in the order of MHz (or even GHz if not based on a cavity architecture). The present setup is far from these demands with 7dB of gain and a bandwidth in the few-Hz regime. Clearly, the bandwidth is a general issue in optomechanical-based amplifier schemes, limited by the narrow mechanical linewidth (even if optical broadened), but large gain has been demonstrated in electromechanical setups, e.g. in Ref. 20,21,25 of the manuscript. Hence, so far I am not convinced that the results presented in the manuscript do warrant publication in Nature Communications."

Reply:

It's true, the amplification performance in regards of bandwidth and gain of the current device is not reaching the values of state-of-the art Josephson parametric amplifiers or the gain reported for other optomechanical amplifiers. This is primarily due to the lower cooperativity that we achieved and the relatively small vacuum coupling rate inherent to the capacitively-coupled nanowire design.

On the other hand, the use of a nanowire, capacitively coupled via the input port of the cavity, did enable us to demonstrate a novel regime not demonstrated before in which (electrostatic) mechanical parametric coupling was able to generate optomechanically-mediated microwave gain.

Finally, we believe that if this same concept were applied with voltage-biased, capacitively coupled devices that can be made at places like NIST, the proof-of-principle experiment we demonstrate here could indeed provide also an improvement in microwave amplification beyond the state of the art, and enable the exploration of novel regimes of cavity electrodynamics such as the demonstration of a negative effective density of states.

In summary: we agree that the performance does not exceed optomechanical conventional amplifiers, but the novelty lies in the new concept of mechanical parametric pumping of an optomechanical cavity.

In addition we would like to add here:

- The actual gain of the intracavity field that we observe is about 13 dB and if we allow for nonlinear behaviour of the mechanical oscillator we even get 19 dB, cf. the new Supplementary Material Sec. 12. The conversion from intracavity gain to net gain could be easily compensated by a slightly larger cooperativity. For example by just having the external cavity linewidth 1 MHz instead of 5.7 MHz (a very simple and realistic

adjustment), we would get a cooperativity of 3 and about 6 dB of additional net gain without any further adjustments.

- We added a new Supplementary Material Sec. 13 and a corresponding discussion to the main paper, where we elaborate on possible improvements to the current sample design. The suggested (simple) improvements should allow for near quantum-limited operation with significantly larger bandwidth and significantly larger gain compared to the existing device.

Reviewer #2:

"Further comments are:

1. The abstract states: *In contrast to many other microwave amplification schemes using electromechanical circuits, the presented technique allows for simultaneous cooling of the mechanical element, which potentially enables this type of optomechanical microwave amplifier to be quantum-limited. First of all 'In contrast many microwave amplification schemes...' is a bit of an overstatement here, clearly, the single-(blue)tone amplification setup in Ref. 21 could not suppress the noise coming from the mechanical mode, but the majority of the recent electromechanical amplifier schemes was based on multi-tone driving with the effect of suppressing the mechanical noise, see Ref. 20, 25 of the manuscript and Phys. Rev. Lett. 118, 103601 (2017). This should be re-phrased and set into the right context."*

Reply:

Yes, indeed, we generalized here a bit too much and we adjusted this part in the revised version of the manuscript accordingly. Due to the limited space in the abstract, we removed the statement there, we do not claim anymore that "many" previously realized optomechanical amplifiers suffer from mechanical noise and we added a discussion to the end of the results section.

Reviewer #2:

"Secondly, the authors speak of the possibility of quantum-limitedness of such an amplifier, but they do not perform an actual noise analysis in their manuscript. And the authors speak of back-action cooling in the conclusion without further elaboration. In the ideal case the results of an actual noise measurement would be desirable, but I can understand that this is not straightforwardly done, however, at least a thorough discussion about the noise properties could be included. In terms of the noise analysis it would be good to understand the limitations coming from the fact that the present setup is not in the resolved sideband regime, in contrast to the proposal in Ref. 26."

Reply:

Clearly, our device is not quantum-limited as we cannot cool the mechanical oscillator into the ground-state nor can we enter the radiation-pressure shot noise limit (i.e. the quantum cooperativity regime). This is not only because we are not in the sideband-resolved limit, but also because our highest cooperativity is 0.5. Therefore, we clearly speak only of the possibility of quantum-limited amplification for an optimized device with reference to Ref. [26], where the noise analysis is done very thoroughly. We do not see how we can contribute to the analysis there any further in a significant way.

We agree, however, that the fact that our device is not in the resolved-sideband regime poses a limit to the minimum achievable phonon occupancy, even assuming high cooperativity and perfect cooling conditions (about 1 residual thermal phonon). We added a statement regarding this point to the manuscript.

Reviewer #2:

"2. As mentioned above the properties of the amplifier are not convincing in terms of gain and bandwidth. The authors should state this more clearly, set their device into context with other cavity-based amplifiers, and include ideas how to improve their design in the future. In addition, it would be interesting to know why it was not possible to achieve a higher gain value."

Reply:

We complied with this request in the revised version with several new statements, discussions and a Supplementary Note.

Reviewer #2:

"3. Otherwise, I mainly have some minor comments:

- Caption Fig.4: *The schematic shown in **d** visualizes the amplification mechanism*, I guess the authors mean the schematic in **a**?
- Typo P.7 last sentence: *mechnaical*."

Reply:

We thank the reviewer for spotting these typos and corrected them in the revised version.

Reviewer #2:

"• General comment: the SI are a bit sketchy and not well-rounded formulated, the authors should consider improving these important additional information."

Reply:

We have worked to improve the language of the Supplementary Material in the revised version.

Reply to reviewer #3:

Reviewer #3:

"In this manuscript, the authors report the parametric amplification of a mechanical beam within a microwave nano optomechanical setup via direct electrical modulation of the mechanical spring constant. They provide evidences by measuring the mechanical gain and the mechanical noise spectrum. They finally show how the parametrically amplified mechanical oscillator allows the amplification of a microwave probe beam in an OMIT setup.

The experiment described and the results shown in the manuscript are of great quality and definitively amongst the state of the art. Showing direct parametric amplification of a mechanical quadrature in a nano optomechanical setup is also of great relevance as many schemes have been proposed in this direction and such results have the potential to motivate more ideas along those lines. While mechanical parametric amplification (and even squeezing below the ZPF which is definitively more challenging) has previously been reported, the present setup is, to my knowledge, novel. Finally, the results are convincing and successfully support their conclusions."

Reply:

We thank the reviewer for his very positive assessment of our manuscript, his appreciation of our results, and his constructive suggestions for improving both.

Reviewer #3:

"I would make two minor suggestions about the manuscript:

1) As mentionner above, I believe in the novelty of the experiment. However I feel that the authors have not given enough credits to previous works where parametric amplification of mechanical elements have been reported. Also, in this context, I would suggest to the authors to point out explicitly the novelty of their work compared to what have been done before."

Reply:

We agree that we did not give enough credit to previous work on parametric mechanical amplification and noise squeezing and added new statements with corresponding references to the manuscript to account for this.

Reviewer #3:

"2) One of the key aspects of amplification is the amount of total noise added to the amplified mode. As shown in the manuscript, the thermal noise is parametrically amplified such that one quadrature is amplified while the other is de-amplified. However, nothing is said about the total added noise (when integrating both quadratures). I believe this is an important quantity to be discussed in order to know if the amplification process is efficient or not. I would suggest to compare the total added noise measured from Fig. 3 to what is expected from an ideal parametric amplifier (which I believe is $n_{amp} = (n_m + 1/2) * \cosh(2r)$ with r the squeezing parameter)."

Reply:

We thank the reviewer for this question, it has stimulated us to perform further analysis of the thermal noise squeezing data.

In particular, we now include in the Supplementary Material Sec. S9 plots showing the power spectral density of both quadratures ($\cos(\omega t)$ and $\sin(\omega t)$) in addition to the full complex quadrature signal ($\cos(\omega t) + i\sin(\omega t)$) shown in the main paper Fig. 3a, so that the output noise spectrum of both the amplified quadrature and de-amplified (squeezed) quadrature can be clearly seen.

We do feel it is an important question if our parametric amplification is adding more noise in the output spectrum than would be expected for an ideal parametric amplification process. To quantify this, we perform an analysis and modelling of the quadrature power spectral densities as well as of

the quadrature variances and find good agreement with theoretical predictions, when taking quadrature-dependent mechanical heating into account.

We believe that this heating in our device appears due to mechanical resonance frequency fluctuations and can be reduced by a higher stability of the mechanical oscillator. Another way to reduce the impact of mechanical frequency fluctuations could be to operate with large optical damping rates, as would be desirable for an optimized performance of the microwave amplification in any case.

Reviewer #3:

“Considering that those two minor points can be easily addressed by the authors, I recommend the manuscript for publication in Nature Communications.

Best,
Marc-Antoine Lemonde”

REVIEWERS' COMMENTS:

Reviewer #1 (Remarks to the Author):

The authors have satisfactorily revised their manuscript, which I would recommend for publication in Nature Communications.

Aurelien Dantan

Reviewer #2 (Remarks to the Author):

The authors have adequately addressed the other referees and my comments. Thus, I think that the manuscript should be published in Nature Comm.

One minor suggestion:

- one could use the standard notation for amplitude gain as \sqrt{G} to avoid confusions.

Reviewer #3 (Remarks to the Author):

I reviewed the corrected manuscript and agree that it is now considerably improved. The work is presented in a more detailed context and its weaknesses/suggestions for improvement are better presented.

I reaffirm my support for its publication.

Reply to the reviewer comments:

Reply to reviewer #1:

Reviewer #1:

"The authors have satisfactorily revised their manuscript, which I would recommend for publication in Nature Communications.

Aurelien Dantan"

Reply:

We are happy to hear that and thank the reviewer for his positive feedback.

Reply to reviewer #2:

Reviewer #2:

"The authors have adequately addressed the other referees and my comments. Thus, I think that the manuscript should be published in Nature Comm."

Reply:

We thank the reviewer for their positive assessment of our revised manuscript.

Reviewer #2:

"One minor suggestion:
- one could use the standard notation for amplitude gain as \sqrt{G} to avoid confusions."

Reply:

We agree that unfortunately there is some confusion in literature. In electromagnetic systems, the photon number gain is usually called G , while in the seminal paper by Rugar and Grütter [34] regarding parametric amplification and noise squeezing in a mechanical resonator G has been used for amplitude gain. As we build our formalism of mechanical amplification, which uses G , on Ref. [34], we prefer to be consistent with the choice there. In the mathematical description of the microwave amplification on the other hand, we do not use the variable G at all.

Reply to reviewer #3:

Reviewer #3:

"I reviewed the corrected manuscript and agree that it is now considerably improved. The work is presented in a more detailed context and its weaknesses/suggestions for improvement are better presented.

I reaffirm my support for its publication."

Reply:

We thank the reviewer for his appreciation of our revision and his reaffirmed recommendation.